bioengineering/biomedical engineering/light microscopy

correlative imaging, histology, blood vessel networks, SIFT, registration, warping

**Author for correspondence:**
Matthew J. Lawson
e-mail: m.lawson@soton.ac.uk

# Immunofluorescence-guided segmentation of three-dimensional features in micro-computed tomography datasets of human lung tissue

Matthew J. Lawson[1], Orestis L. Katsamenis[2], David Chatelet[1], Aiman Alzetani[1], Oliver Larkin[3], Ian Haig[4], Peter Lackie[1], Jane Warner[1] and Philipp Schneider[3,5]

[1]School of Clinical and Experimental Sciences, Faculty of Medicine, [2]μ-VIS X-ray Imaging Centre, Faculty of Engineering and Physical Sciences, and [3]Bioengineering Research Group, Faculty of Engineering and Physical Sciences, University of Southampton, Southampton, UK
[4]Nikon X-Tek Systems Ltd, Tring, UK
[5]High-Performance Vision Systems, Center for Vision, Automation and Control, AIT Austrian Institute of Technology, Vienna, Austria

MJL, 0000-0003-0115-1698; AA, 0000-0002-3373-6714; OL, 0000-0003-4799-9563; PS, 0000-0001-7499-3576

Micro-computed tomography (μCT) provides non-destructive three-dimensional (3D) imaging of soft tissue microstructures. Specific features in μCT images can be identified using correlated two-dimensional (2D) histology images allowing manual segmentation. However, this is very time-consuming and requires specialist knowledge of the tissue and imaging modalities involved. Using a custom-designed μCT system optimized for imaging unstained formalin-fixed paraffin-embedded soft tissues, we imaged human lung tissue at isotropic voxel sizes less than 10 μm. Tissue sections were stained with haematoxylin and eosin or cytokeratin 18 in columnar airway epithelial cells using immunofluorescence (IF), as an exemplar of this workflow. Novel utilization of tissue autofluorescence allowed automatic alignment of 2D microscopy images to the 3D μCT data using scripted co-registration and automated image warping algorithms. Warped IF images, which were accurately aligned with the

µCT datasets, allowed 3D segmentation of immunoreactive tissue microstructures in the human lung. Blood vessels were segmented semi-automatically using the co-registered µCT datasets. Correlating 2D IF and 3D µCT data enables accurate identification, localization and segmentation of features in fixed soft lung tissue. Our novel correlative imaging workflow provides faster and more automated 3D segmentation of µCT datasets. This is applicable to the huge range of formalin-fixed paraffin-embedded tissues held in biobanks and archives.

## 1. Introduction

The respiratory system comprises a complex branching network of airways and blood vessels ranging in size down to the micrometre scale. X-ray microtomography provides non-destructive three-dimensional (3D) imaging [1], which is also commonly known as X-ray microfocus computed tomography, micro-computed tomography and micro-CT (µCT); all of these are correct and acceptable terms but to follow with the most recent literature [2–5] it shall be referred to as µCT in this paper. Some modern µCT systems have been optimized for high-contrast imaging of soft tissue biopsies (e.g. lung) prepared as standard formalin-fixed paraffin-embedded (FFPE) blocks. This technique of 3D imaging of lung tissue can be achieved at a spatial resolution of the order of 10 µm without the need for contrast agents, which would complicate the use of the tissue for additional imaging modalities [6] and is known as 3D X-ray histology (XRH) [2]. µCT tissue volumes are often visualized as a series of hundreds or thousands of two-dimensional (2D) images with microstructural details comparable to destructive 2D histological tissue sections. Correlation of non-destructive 3D µCT data with destructive 2D wide-field histology images of paraffin wax sections, from the same lung tissue block, allows the identification and 3D localization of specific tissue components and cell types [7].

Segmentation is the process of identifying and labelling a specific feature or area of interest from the rest of an image [8]. In the case of histology and µCT images, segmentation is most often a manually intensive process, potentially subjective as the user identifies and labels specific features on hundreds of µCT images using the histology as a reference [9]. As a result, image segmentation of specific features can take days or weeks to complete if hundreds or thousands of reconstructed µCT slices are used [8]. To date, the majority of such correlative studies have employed bright-field microscopy to localize features seen in histologically stained tissue sections or immunohistochemistry (IHC) for µCT image segmentation. Immunostaining of tissue provides the localization of specific immunoreactivity and has most commonly been visualized by bright-field chromogen-based staining (IHC); however, this can be difficult to segment due to low image contrast and variable image colour. Examples of correlative lung studies with quantitative analysis of manual techniques have been used to investigate lung lymphatics [5], airways in chronic obstructive pulmonary disease (COPD) [10] and fibroblasts in idiopathic pulmonary fibrosis [11]. The next logical step for future use of XRH is to incorporate more automated means of segmenting immunostaining of cells and features in 3D µCT datasets. This would enable studies like those listed previously to be completed in less time as well as permitting a higher throughput of samples to be imaged and analysed in order to derive biologically significant conclusions from correlative imaging data. Immunofluorescence (IF) staining, less commonly used for FFPE tissue and not previously used for correlative µCT studies of soft tissues, can generate higher contrast microscopy images compared to IHC. The distribution of immunolabelled areas in the images, identified by high IF image contrast, could thus be easily extracted by thresholding based on image intensity [12]. Automating steps for segmentation using IF imaging could reduce the time taken from several days/weeks with current fully manual techniques, as well as removing subjective manual inputs by saving the user from manually identifying and segmenting features from µCT images.

Direct image registration of µCT and 2D histology images could greatly improve the resulting correlation and segmentation of the images involved. However, when FFPE tissue is sectioned the tissue can be stretched, compressed, torn or even lost [13]. This compromises direct (i.e. rigid or inelastic) image co-registration between the 2D microscopy images and the corresponding plane in the µCT data [7]. Existing registration tools including more applicable deformable (elastic) methods have been reviewed by Ferrante & Paragios [14]. However, many of the image registration methods reported in this review, as well as available registration tools in open source (e.g. ImageJ [15] and ec-CLEM [16]) and commercial software (e.g. Avizo (FEI) and ZEN/ATLAS (Zeiss)), require high amounts of manual user input to identify matching features in the images which can take several

hours or more to complete; this remains a major bottleneck in the processing of these types of images. Automatically locating matching features between µCT and histology images, which take into account the heterogeneous nature of lung tissue or heavily diseased tissue, requires further development to automate the registration process.

Segmentation of µCT images using immunostaining for reference is currently possible but is an almost entirely manual process which requires large amounts of time (several weeks/months) and effort to do [5]. This work developed techniques of automated registration of fluorescent images to µCT to produce a correlative imaging workflow not used in previous studies. The goal of this was to reduce total image processing time and manual interactions with the data in order to produce a workflow which could be used by the wider community to perform higher-throughput correlative imaging studies of biological tissues, like those found in biobanks, using µCT and IF. The airway epithelium and blood vessel networks of the lungs were used as exemplars to develop the segmentation workflow. This work for the first time brings together, in a consistent and reproducible way, structural and functional imaging-based characterization of tissues using 2D IF and 3D µCT.

# 2. Material and methods

## 2.1. Correlative human lung tissue imaging

Human lung tissue was obtained from patients undergoing resection surgery at the University Hospital Southampton. Patients gave signed informed consent prior to surgery and ethical approval was provided by the Southampton and South West Ethics Committee (number 08/H0502/32). Two exemplar surgical lung tissue samples, for method development (approx. 1–2 cm cubes), were selected from macroscopically normal peripheral lung tissue. These were fixed in neutral buffered formalin for 48 h, embedded in paraffin wax and mounted onto a plastic cassette following standard histopathological protocols [17] (step 1 in figure 1).

Prior to imaging by µCT, the plastic cassette was removed from the wax tissue blocks and excess paraffin wax was trimmed away using a scalpel. All samples were scanned using a custom-designed µCT system optimized for unstained soft tissues (Med-X prototype; Nikon X-Tek Systems Ltd, Tring, UK) [2]. The µCT scanner was operated at an X-ray tube potential (peak) of 55 kVp, the beam current being set to 125 µA at a power of 6.9 W using a static reflection X-ray target. Voxel spacing ranged between 6.0 and 8.5 µm depending on tissue size, at an exposure time of 1.25 s per projection using 4001 projections and four frames per projection. The total µCT scan time per sample was approximately 8 h. The data were reconstructed, using conventional filtered back projection, as 32-bit floating-point volumes using proprietary CT reconstruction software (CTPro version V5.1.6054.18526; Nikon X-Tek Systems Ltd). Technical details of the µCT scanning protocol were based on those of Scott [6] and further optimized by Katsamenis *et al.* [2]. Following CT reconstruction, the data volumes were converted to 16-bit and manually cropped to the boundaries of the tissue within the scan using the open-source image processing and analysis package Fiji [15].

Following non-destructive µCT imaging, 100 serial sections (4 µm thickness) were cut from the same FFPE tissue block using a Leica RM2135 wax microtome (Leica Biosystems, Germany); this volume of serially sectioned tissue was dubbed the 'sectioned volume'. Three sections were stained with Mayer's haematoxylin and eosin (H&E) following standard histological protocols [18]. Sections were stained for IF using a single set of primary and secondary antibodies following the method detailed by Robertson & Isacke [19] for staining FFPE tissue. Cytokeratin 18 (Ck18) was chosen as the exemplar primary antibody (C8541; Sigma-Aldrich, USA) to immunolabel Ck18 positive columnar epithelial cells of the airways. Ck18 staining was visualized using an Alexa Fluor 647-conjugated goat anti-mouse IgG secondary antibody (Invitrogen A21235; Fisher Scientific, USA), to provide an imaging wavelength with the least tissue autofluorescence signal. 2D wide-field images of whole tissue sections were captured and digitized with a fluorescence slide scanner (VS110; Olympus, Japan) at a nominal resolution of 0.645 µm/pixel using a 10× objective with a numerical aperture of 0.4. Fluorescence images were acquired using metal halide arc excitation (X-cite Exacte 200w) through a fast filter wheel with a DAPI/FITC/CY3/CY5 hard-coated and high-transmission filter set (filters: 377/50 HC, 485/20 HC, 560/25 HC and 650/13 HC). Emission was collected through a matched, quadruple band, polychroic mirror and emission set and captured on an Olympus XM10

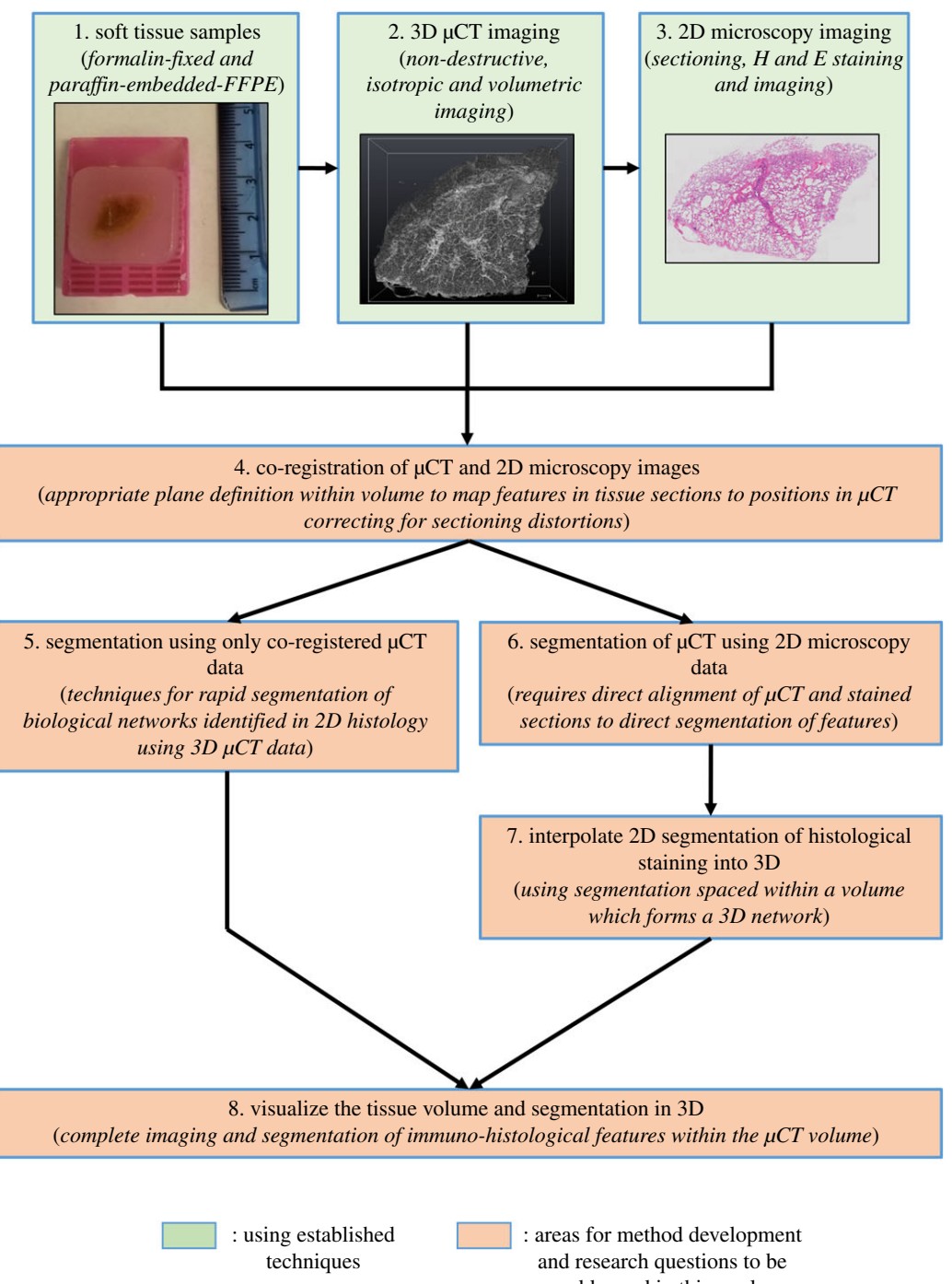

**Figure 1.** Summary of the proposed correlative imaging workflow highlighting the research question addressed in this work. Steps highlighted in green have been previously published. The proposed steps highlighted in orange were identified as areas for the development of greater automation. 1: Soft tissue sample preparation via FFPE. 2: Imaging of lung tissue at a high resolution in 3D using μCT. 3: Sectioning of tissue and employing traditional 2D histological imaging techniques to identify specific features not visible in μCT imaging. 4: The first step to address by co-registering the data from steps 2 and 3. 5: Segmentation of 3D networks using the μCT volume while avoiding manual segmentation. 6: Localization of specific features identified by 2D histology within μCT without manual segmentation. 7: Building non-isotropic segmentation of section staining into 3D isotropic data. 8: Visualizing the results of the previous steps together in 3D.

monochrome camera. All acquired images were background corrected against reference images captured with the fluorescence shutter closed. The fluorescence images were captured using two of the available channels at excitation/emission wavelengths of 490/520 nm (FITC) at an exposure time of 1000 ms for autofluorescence and 650/680 nm (CY5) at an exposure time of 250 ms for the IF staining.

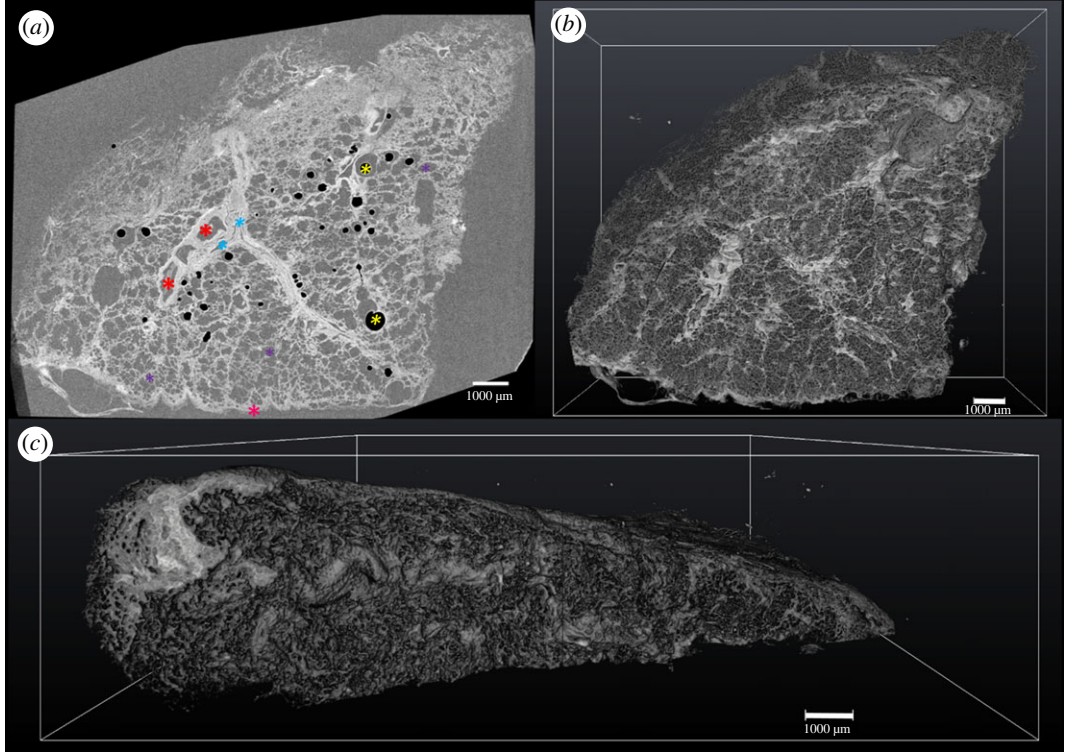

**Figure 2.** Reconstructed µCT data in 2D and 3D of FFPE human lung tissue sample. (*a*) Reconstructed µCT slice in the *xy*-plane of the µCT volume. Examples of key features of the lung tissue are labelled: airways (blue *), blood vessels (red *), alveoli (purple *), pleural surface (pink *), masked air bubble artefacts (yellow *). (*b*) 3D volume rendering of the lung tissue volume in the *xy*-plane. (*c*) 3D volume rendering as in (*b*), in the orthogonal *yz*-plane. (*a–c*) Voxel size of µCT scan, 8.5 µm; scale bars, 1 mm; tissue from patient 1.

## 2.2. Image processing and µCT segmentation of blood vessels

The 16-bit µCT data were firstly resampled in Fiji, to orient the *xy*-plane to approximately match the sectioned face of the tissue block, by rotating and reslicing the 3D dataset using bicubic interpolation (see electronic supplementary material, figure S1, for a pictorial representation of the planes involved in handling 3D tissue data). Based on images from 2D H&E stained sections, the 3D µCT volume was rigidly resliced to align with the plane of histology sectioning using the 3-point alignment (slice module) in the commercial image processing and visualization software package Avizo (v. 9.3; Thermo Fisher Scientific) as described by Robinson *et al*. [5]. This allowed co-registration (step 4 in figure 1) of the 2D thin-section histology image (step 3 in figure 1) and the corresponding plane within the µCT data (step 2 in figure 1). µCT data of lung tissue volumes containing air bubbles, due to insufficient wax infiltration like those seen in figure 2, were processed to remove the edge enhancement which outlined the air bubbles to prevent them from showing as features in the segmentation and visualization of the µCT dataset. Using the WEKA trainable segmentation 3D tool in Fiji [20] a classifier was trained (electronic supplementary material, file S1), using two classifiers with default feature settings, on µCT datasets containing air bubbles. This was used to automatically segment the air bubble voids and the bright bands surrounding the air bubbles throughout the volume. The inverted segmentation images of the air bubbles had the grey value of the bubble and edge-enhanced outlines set to zero and the rest of the image set to 1. This 'mask image' was multiplied by the original µCT image to set the grey value of the bright bands and air bubbles to zero, while leaving the rest of the image unchanged to facilitate visualization and segmentation of the µCT volume.

To segment blood vessels from the µCT dataset in the least amount of time possible (step 5 in figure 1), we used the 'active contour segmentation' tool in the ITK-SNAP software package (v. 3.6, www.ITKsnap.org) [21]. An adjustable threshold was applied to sub-volumes of the µCT dataset, to minimize computer memory usage, defining the blood vessel boundaries. The lumen was set to the lower threshold pixel values (19 000–24 000) and the walls of the blood vessels set the upper threshold

pixel values (24 001–65 535). Seed points for the 'active contour segmentation' were manually placed ($n = 40 \pm 5$) in the centre of the lumen, using H&E and/or IF images as a reference. The segmentation settings in ITK-SNAP were all kept at the default values except for the 'smoothing force' (resistance for entering narrow objects, default value = 0.2) which was increased to 0.4 to avoid 'leaking' into neighbouring airspaces in smaller vessels. Running the 'active contour evolution' causes the segmentation to grow from the seed point and can be observed in real time in 2D ($xy$, $xz$, $yz$ views) and 3D. This process was repeated on new sub-volumes of interest until the whole volume was segmented and exported as an 8-bit segmentation mask image. Due to the computationally intensive processing (high memory demands) of the 'active contour segmentation' the software was run using the IRIDIS 5 visualization cluster (10 core CPU, 192 GB RAM, 8 GB NVidia M60 Tesla GPU) at the University of Southampton.

## 2.3. Novel co-registration of immunofluorescence and μCT data

After the acquisition of the 2D IF images and μCT volume, the data were co-registered in order to address step 4 in figure 1. Due to sectioning artefacts present in the IF images, this required multiple processing steps. First, 2D IF section images were manually rotated and downscaled using bicubic interpolation in Fiji [15] to match the orientation and size of the corresponding reconstructed 2D μCT slices. The autofluorescence images (excited at 495 nm) were digitally warped, using an affine transformation, to fixed reference images provided by corresponding μCT planes free of sectioning artefacts. Automatic feature extraction tools in Fiji were combined with the BigWarp plugin in Fiji, which uses landmark-based deformable image alignment using thin-plate splines [22] and is found within the BigDataViewer [23], to create the 'automated warping script' (electronic supplementary material, file S2). The workflow of the script processes can be visualized in electronic supplementary material, figure S2. The script opens the first matching pair of autofluorescence and μCT images in the input directory, based on the user-defined file nomenclature ensuring matched images are opened for registration (step 2 in electronic supplementary material, figure S2). Multiple scale invariant feature transform (SIFT) correspondence algorithms [24] are applied to the images with different size filters. This provides a hierarchy of feature sizes to optimize the matching of feature points between the autofluorescence and μCT images. The points' coordinates ($xy$ location in each 2D image) were saved and used to apply the BigWarp transformation on the autofluorescence channel image, followed by transformation of the Ck18 IF channel image using the same landmarks. This whole process was repeated for each pair of corresponding fluorescence and μCT images in the input directory producing a series of warped IF images. Following automated registration and warping the process was repeated with manually placed landmarks, between matching histological features visible in both images, using the BigWarp plugin for comparison to automated landmarks generated with the script.

The resulting co-registered images were assessed for the proportion of 'accurate registration'. This was defined as tissue overlapping with tissue, or airspaces overlapping with airspaces in the fluorescence and μCT images. 'Inaccurate registration' was defined as areas where airspace overlapped with tissue or *vice versa*. The two matching registered images were thresholded as an 8-bit mask by selecting the tissue (pixel value 255) and excluding the airspaces (pixel value set to 1 after thresholding). These images were multiplied together and any pixels with a value of 255 were designated inaccurate registration with the remainder being designated accurate registration. The amount of registration inaccuracy was visualized by applying a colour map to the resulting image. This was plotted on a bar-chart showing the combined areas of accurate registration (blue and cyan in the images) as one blue column and the inaccurate as a red column; this analysis was completed on a series of 22 pairs of images throughout the sectioned volume of tissue.

## 2.4. Segmentation of registered immunofluorescence images

To address step 6 in figure 1, an absolute threshold was applied to the warped IF images which created binary segmentation masks of Ck18 positive staining registered to each corresponding μCT slice. A new 8-bit volume of the same dimensions as the μCT volume was then created in Fiji. Each 2D IF segmentation mask of the Ck18 positive staining was inserted in the position (slice number) of the corresponding μCT slice within the sectioned volume (total depth of tissue approx. 400 μm). Not every serial section was stained and imaged for Ck18 localization (sections up to 40 μm apart); therefore, to address step 7 in figure 1, the intermediate μCT slices between the stained sections were digitally interpolated using 'ND morphological contour interpolation' implemented in ITK-SNAP [25]. Interpolation errors were identified as segmentation leaking into airspaces or alveoli (red areas in

electronic supplementary material, figure S4). These were excluded using a binarized μCT image, created by applying an absolute threshold, which selected the tissue and excluded the airspaces (filled with paraffin wax) in the μCT dataset. The interpolated segmentation mask was multiplied by this binary μCT image, removing any segmented voxels in the airspaces or lumen. The masked interpolation was used for 3D analysis and visualization of the Ck18 segmentation. For comparative visualization, and to address step 8 in figure 1, the μCT (as a volume render), blood vessel segmentation and Ck18 segmentation (as 3D surfaces) were visualized in the sectioned volumes of tissue using Avizo.

# 3. Results

Here the results of the μCT and IF imaging are shown, providing structural information in the 3D domain (e.g. blood vessels) while identifying specific immunoreactivity in 2D (e.g. Ck18 immunoreactivity) which could be mapped to the volume. These methods (steps 4–7 in figure 1) decreased the time taken to generate the segmented data, from several weeks of manual work down to several hours (depending on the number of IF images used), and also combined the information provided by 2D IF and 3D μCT.

## 3.1. Correlative human lung tissue imaging

Using the μCT imaging developments of Scott and colleagues and Katsamenis and colleagues [2,6] we successfully imaged FFPE lung tissue to visualize the structural features of the lung at spatial resolutions less than 10 μm. There was sufficient image contrast and signal strength in the μCT between the lung tissue and the surrounding paraffin to identify airways, blood vessels and additional features of the lung tissue (figure 2*a*). The scanned tissue volume contained a series of 400 reconstructed CT slices (800 slices in dataset 2 shown in electronic supplementary material, figure S6), illustrating the biological features mentioned above. The full tissue volume was rendered and visualzsed using orthogonal views to reveal the full structure of the lung tissue within the wax block (figure 2*b*,*c*).

Airway epithelium and specific cell types could not be identified as clearly in the μCT datasets as the blood vessels, mainly due to insufficient image contrast and the cells being smaller than the voxel spacing of the μCT scans (6.0–8.5 μm). However, these were identified and localized by IF staining of tissue sections from the same tissue scanned by μCT using Ck18 as an exemplar primary antibody for cell-type-specific antigens. Using H&E images as a reference for determining the sectioning plane, the μCT was successfully resliced into the orientation matching the plane of the serially sectioned tissue, enabling the IF images to be registered to the μCT data. IF staining identified the location of the Ck18 positive columnar airway epithelium in FFPE lung tissue sections and matched histologically distinct areas of epithelium on the adjacent H&E stained section (figure 3*a*,*b*). Collapsed portions of the airways were commonly seen in both 2D histology and 3D μCT data (Aw in figure 3). Autofluorescence provided a good overview of the tissue histology, matching the gross structure seen by H&E staining (figure 3*a*,*c*). The high image contrast of the IF staining over the background (figure 3*b*) proved valuable in enabling thresholding of the Ck18 staining for image segmentation. Blood vessels and structural tissue were clearly visible in the H&E (figure 3*a*) and the tissue autofluorescence (figure 3*c*) images and so could be used as seed point references for blood vessel segmentation. Combined with the autofluorescence channel the location of the Ck18 staining relative to the tissue parenchyma was imaged in the whole section by wide-field microscopy (figure 3*d*,*e*).

## 3.2. Blood vessel segmentation

The blood vessel walls have relatively high X-ray absorption in relation to the rest of the tissue, due to high levels of collagen, making them easily distinguishable and facilitating threshold-based segmentation using ITK-SNAP. Seed points were added to the dataset and the progress of the segmentation from a single seed point is seen in 2D (figure 4*a*) and 3D (figure 4*b*). The progress of the segmentation after additional seed points were added throughout the volume is shown in 3D (figure 4*b*–*d*), illustrating how the network was 'grown' by adding up to 40 seed points. Blood vessels down to approximately 25 μm diameter were segmented from the whole μCT volumes in 4–5 h. Smaller vessels and capillaries were close to the resolution limit of the μCT scans and so were not segmented. Blood vessel segmentation of the second lung tissue sample (electronic supplementary

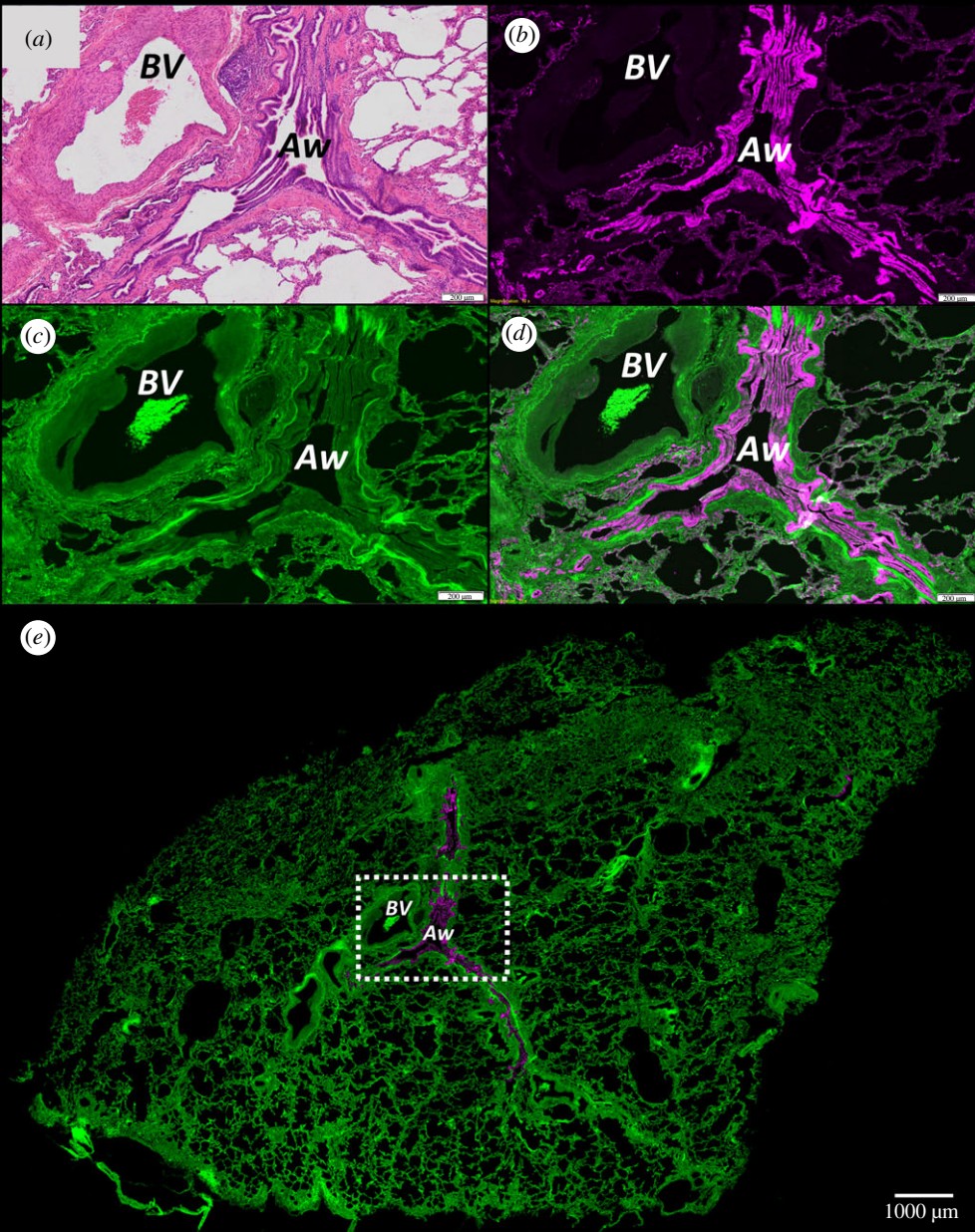

**Figure 3.** Bright-field histological (H&E) and IF staining of human lung tissue sections. (*a*) H&E staining of human lung tissue. (*b*) Ck18 IF staining of the airway epithelium captured at 650 nm, on a neighbouring section approximately 4 μm apart. (*c*) Tissue autofluorescence captured at 480 nm, focused on the same area of tissue as in (*b*). (*d*) Panels (*b*) and (*c*) combined showing the IF channel overlaid on the autofluorescence channel. (*e*) Wide-field microscopy image of whole tissue section comprising the regions of interest shown in (*b*–*d*) within the white box. (*a*–*e*) Images were captured using a 10× objective on an Olympus VS110 slide scanning microscope (Olympus, JP). Airways (Aw) and blood vessels (BV) are labelled. (*a*–*d*) Scale bar, 200 μm; (*e*) scale bar, 1 mm; tissue from patient 1.

material, figure S6) showed that the size of the lung tissue did not greatly affect the number of seed points required nor the time taken to segment.

## 3.3. Image co-registration and warping

To localize specific IF staining within the volume we needed to combine the information from the 2D IF data with the 3D μCT data. However, by overlaying the original autofluorescence image onto the μCT it was possible to visualize the differences between the μCT and the fluorescence images (figure 5*a*). Comparing the location of structural features (e.g. blood vessels) there was a visible mismatch between the images of up to 500 μm. These sectioning artefacts made it impossible to directly localize

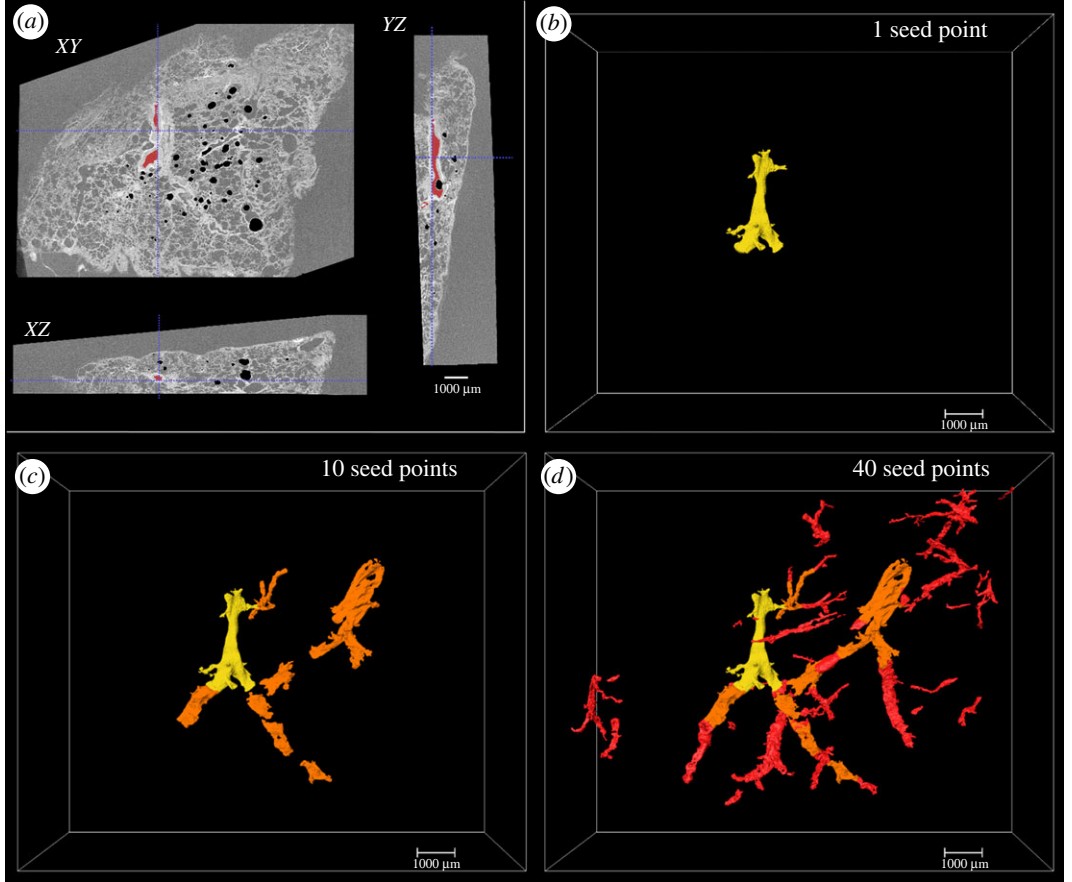

**Figure 4.** Segmentation of the lung blood vessel network using 'active contour segmentation'. (*a*) The three orthogonal CT planes as they appear after initiating the 'active contour segmentation' from a single seed point in ITK-SNAP. (*b–d*) 3D rendering of the blood vessel network as more seed points are added in order to 'grow' the network. Yellow = 1 seed point, orange = 10 seed points, red = 40 seed points; voxel size of μCT scan, 8.5 μm; scale bar, 1 mm; tissue from patient 1.

and segment the immunostaining in the μCT. Planes from the μCT volume and their corresponding autofluorescence images were processed by the 'automated warping script' (electronic supplementary material, file S2) in Fiji. This generated elastically warped fluorescence section images (autofluorescence and IF) registered to the corresponding μCT slice in approximately 2 min. The improvement in image correspondence between the μCT and autofluorescence images after warping was visualized by overlaying the warped autofluorescence image generated by the 'automated warping script' onto the μCT slice (figure 5*b*). This facilitated direct correlation of the information present in the two imaging modalities. Example warped H&E and IF images generated from the 'automated warping script' can be seen in electronic supplementary material, figure S3.

As lung comprises tissue containing clearly distinguishable airspaces, inaccurate registration of fluorescence to μCT images could be clearly identified where airspaces mapped to tissue, or *vice versa* (figure 5*c,d,e*, red areas). By using the BigWarp plugin in Fiji, the proportion of accurately registered pixels corresponding to tissue or air in the autofluorescence and μCT images was increased from approximately 50% in the best rigidly aligned image planes to approximately 74% over a series of 22 images (figure 5*f*). Manually defining the landmarks (approx. 50 per image) and warping the images took a user familiar with BigWarp approximately 20–30 min per pair of images. Using the 'automated warping script' to place (approx. 50–200) landmarks (depending on section size) based on hierarchical SIFT correspondence reduced the time taken to approximately 1–2 min per image pair, with no additional user input or selection required. This variation in the number of automatically generated landmarks had little effect on the percentage of accurate registration being between 65% and 80% in all processed pairs of images. Additionally, the proportion of accurately registered areas was almost identical to the warping produced from the manually defined landmarks (figure 5*f*). Visual inspection also showed a good correspondence with the tissue histology. This illustrated similar levels of registration between manual user-defined registration and the automated approach presented here.

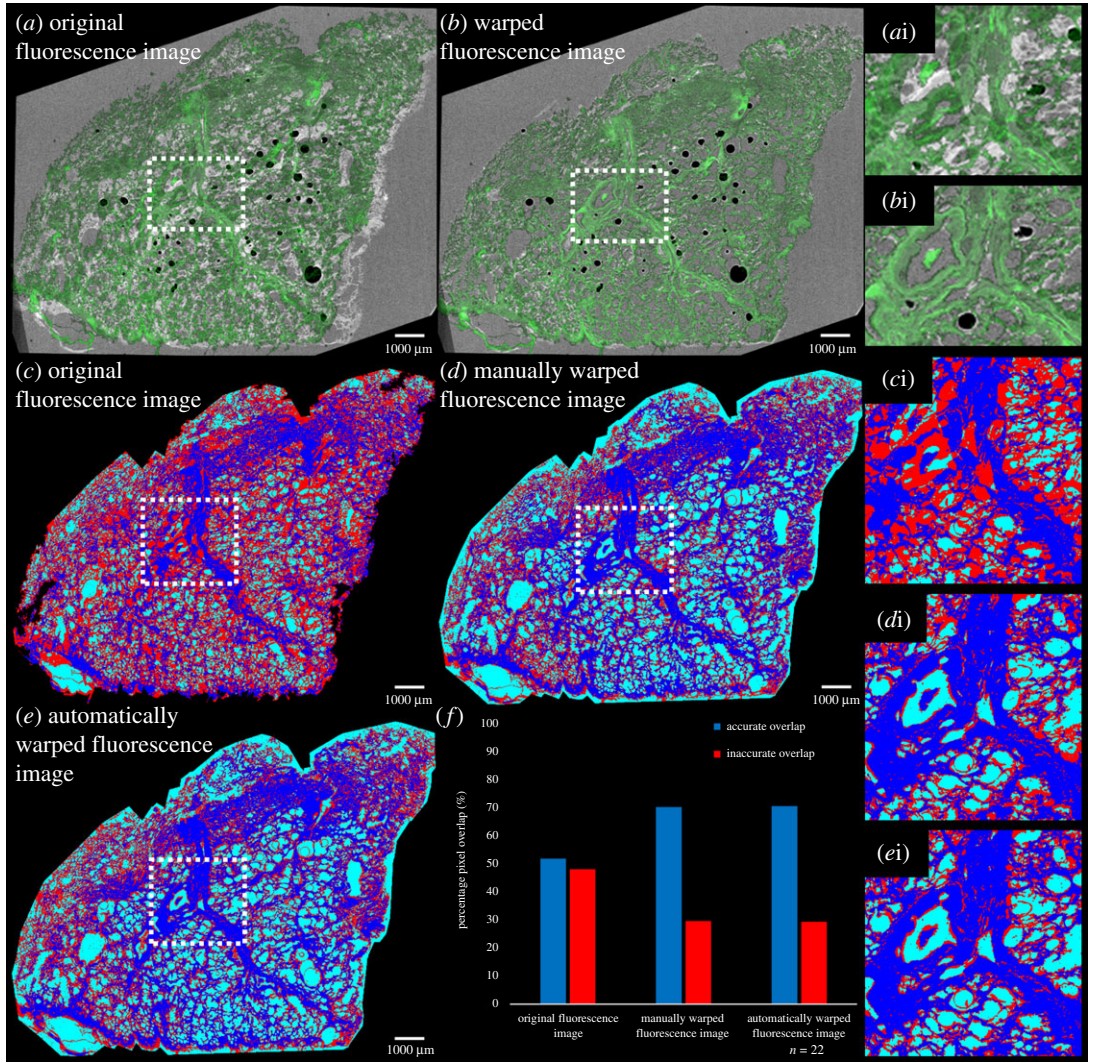

**Figure 5.** Assessment of fluorescence and μCT image co-registration. Images illustrating feature match of tissue and airspaces of the original and warped autofluorescence section with the matching plane from the μCT dataset. (*a*) Original autofluorescence section (green) overlaid on corresponding μCT plane (grey). (*b*) Warped fluorescence section overlaid on corresponding μCT plane. Inaccurate (tissue–air) overlap shown in red, accurate overlap in cyan (airspace–airspace) or blue (tissue–tissue). (*c*) Original autofluorescence section (as in *a*). (*d*) Original autofluorescence section with manually defined warping. (*e*) Original autofluorescence section output from using the 'automated warping script'. (*a*i–*e*i) Corresponding magnified images of areas in the regions of interest highlighted in (*a*–*c*). (*f*) Bar-chart showing the percentage of pixels with accurate (cyan and blue combined) and inaccurate overlap (red) in (*a*–*c*); for the automatically warped fluorescence images the mean ± s.d. is shown (*n* = 22 pairs of images). Voxel size of μCT scan, 8.5 μm; scale bars, 1 mm; tissue from patient 1.

## 3.4. Immunofluorescence-based segmentation

Specifically localizing staining from IF and using that to segment the μCT volume was achieved, with minimal user input, using the direct registration and warping of the fluorescence images to the μCT. The registered IF staining highlighted the location of the Ck18 positive staining without the need for any forms of manual segmentation (figure 6). Ck18 was chosen because the airways are visible in both the μCT and IF images, so that we could easily discern whether the image registration was accurate by looking at the tissue morphology visible in both images. Multiple μCT slices with IF-based segmentation from Ck18 immunostaining were generated showing accurate matching with no major offset from the airways where Ck18 is located. The blood vessels contained no Ck18 staining (figure 6) and there was no cross over with the blood vessel segmentation previously described.

Non-neighbouring paraffin wax sections were immunostained and segmented (figure 6) for Ck18; therefore, the intermediate 'gaps' between the sections needed to be filled. This was achieved successfully using 'ND morphological contour interpolation' [25] in ITK-SNAP. A binarized version of

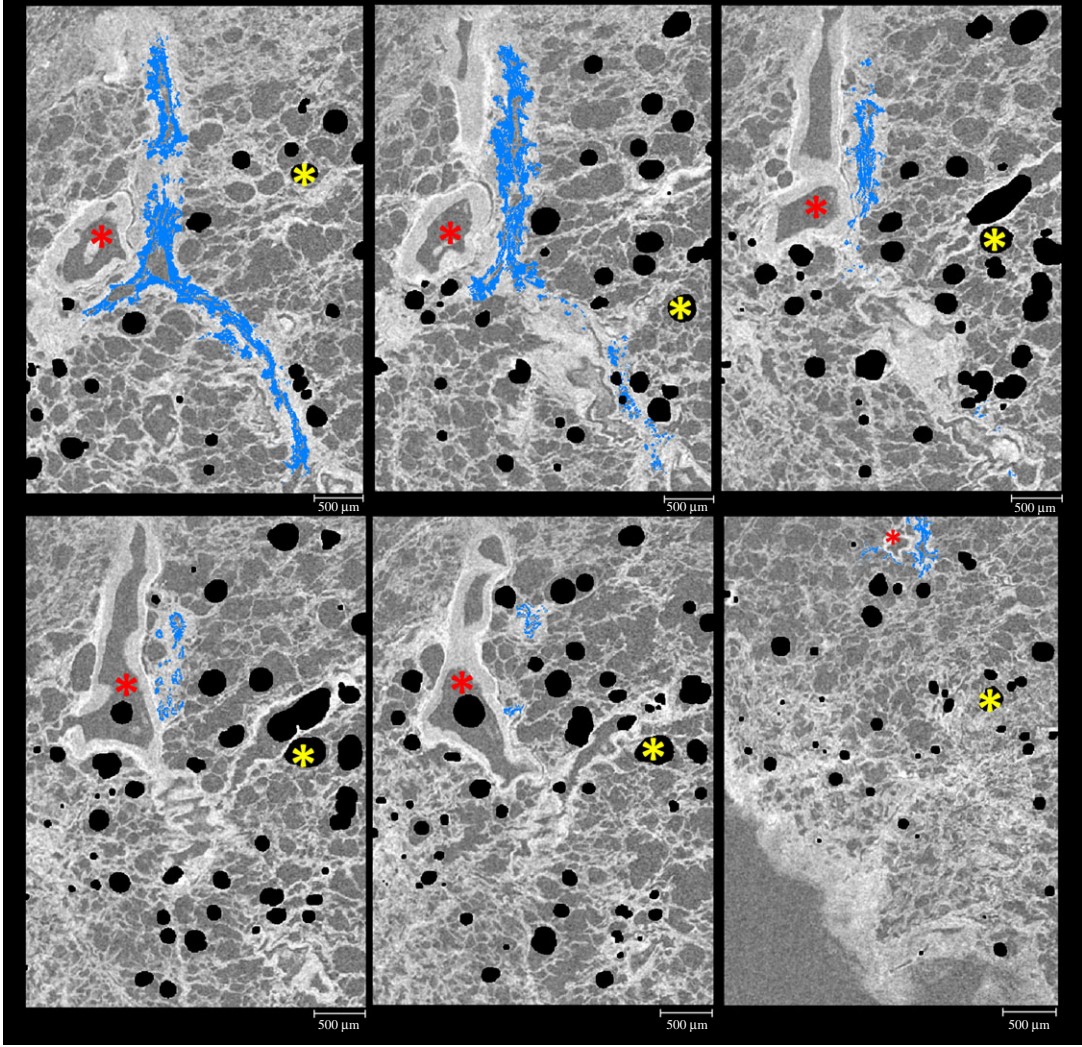

**Figure 6.** Identification of the Ck18 positive staining in µCT data. Ck18 positive staining of the epithelium (blue) identified on six IF stained tissue sections which have been co-registered and segmented on the µCT slices down through the lung volume. The staining location within the tissue was taken directly from the IF images, like those displayed in figure 2. Ck18 positive areas are not present in the blood vessels (red *). These panels follow the airway over a depth of approximately 400 µm. Air bubbles artefacts (yellow *); voxel size of µCT scan, 8.5 µm; scale bars, 500 µm; tissue from patient 1.

the µCT slice was successfully used to correct any interpolation errors in intermediate images (electronic supplementary material, figure S4). The result of this was a 3D network of the Ck18 localization, within the sectioned volume, without the need for manual segmentation.

## 3.5. Three-dimensional visualization of blood vessel and cytokeratin 18 segmentation

The 3D visualizations of the µCT volume containing the segmented networks of blood vessel and Ck18 positive staining can be seen in figure 7. A summary of all the visualization of exemplar lung tissue 1 and 2 can be found in electronic supplementary material, figures S5 and S6. The blood vessels formed complex 3D networks with high levels of interconnectivity throughout the tissue volumes. The volume of the blood vessel segmentation was calculated in Fiji and showed that tissue 1 had a sectioned network volume of 1.34 mm$^3$ and tissue 2 had a sectioned network volume of 5.78 mm$^3$ (down to a minimum detectable thickness of 25 µm). The interpolated Ck18 localization was also visualized within the sectioned µCT volume, thus providing inferred 3D datasets of the Ck18 positive cell localization, and showing the columnar epithelial cells of the airway epithelium within the lung tissue volume. In this view (figure 7), when combined with segmented blood vessels, we can observe that there are fewer Ck18 positively stained areas of the tissue than identified blood vessels. Indeed, the sectioned volume of Ck18 positive staining represents 0.19 mm$^3$ in tissue 1 and 0.50 mm$^3$ in tissue

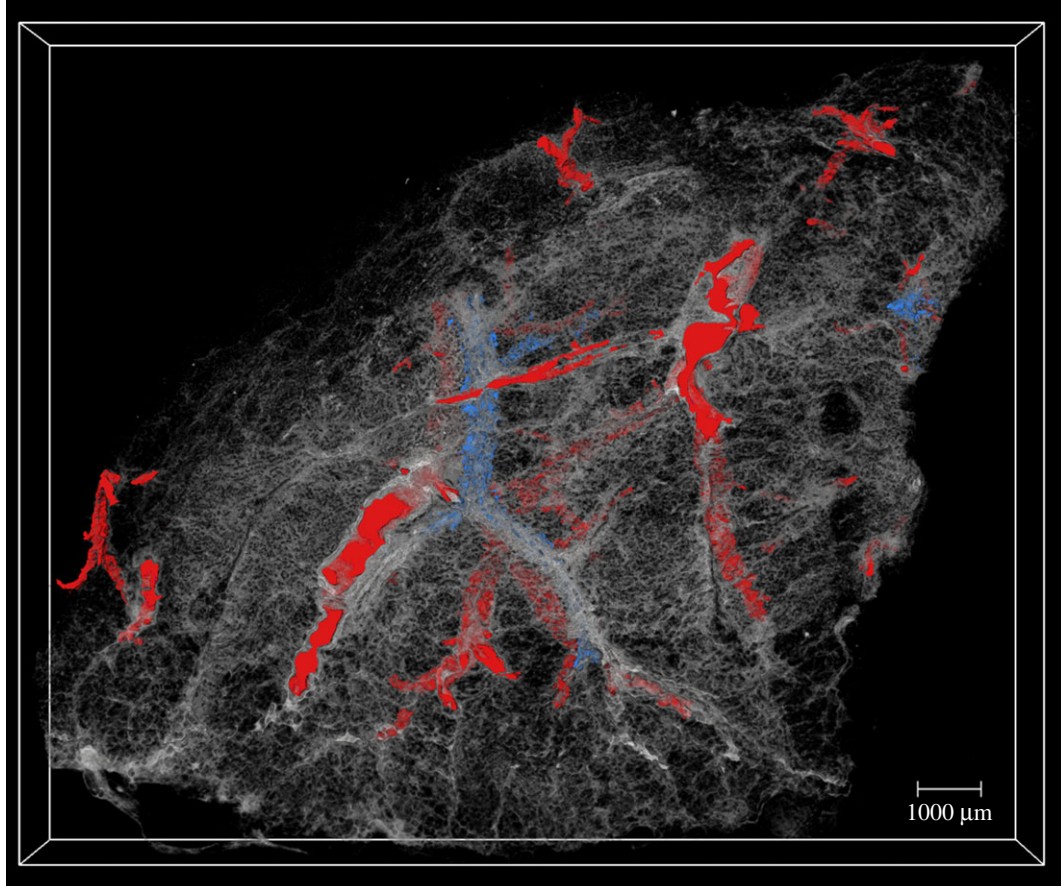

**Figure 7.** Volume rendering of the 3D μCT data containing the segmentation results of the blood vessels and Ck18 staining. The μCT data (as shown in figure 2) were made semi-transparent and cropped into the physically sectioned volume in order to visualize the internal networks of Ck18 and blood vessels. In red, the complete blood vessel network from figure 3 is shown. The interpolated Ck18 staining is shown in blue, which has been generated from the combined results of IF imaging and co-registration shown in figures 4–6.

2, compared to 1.34 mm$^3$ and 5.78 mm$^3$ for the blood vessel in tissues 1 and 2, respectively. The μCT, blood vessel network and Ck18 segmentation in the sectioned volume of both datasets can be followed in 3D by interactively scrolling through the 2D slices in the dataset and viewed in 3D as shown in electronic supplementary material, videos S1 and S2.

## 4. Discussion

Using 3D μCT image datasets of soft tissue has the potential to transform what can be visualized in 3D at micrometre resolution, just as patient CT provides insightful 3D information at millimetre resolution. Proof of concept was demonstrated by Scott *et al.* [6] and the resultant μCT images at typical isotropic voxel spacing of 1–10 μm [2] are sufficient to reveal, in 3D, the network of small airways (less than 2 mm diameter) and blood vessels. These cannot be fully resolved using standard (*in vivo*) 3D imaging techniques, such as clinical CT (0.5–2 mm spatial resolution) or magnetic resonance imaging (less than or equal to 1 mm spatial resolution). While traditional thin-section histology can provide superior spatial resolution (down to the diffraction limit of visible light at around 0.2 μm), these light microscopy images are inherently 2D. Creating 3D images through the stacking of serial 2D tissue sections is feasible, but technically challenging and time-consuming. The limited scope bears the risk of misinterpreting the resulting 3D datasets [11,26]. Thus XRH greatly facilitates the histological investigation of lung tissue allowing larger 3D network structures and 3D relationships to be explored in relevant tissue volumes.

The physiological functioning of the lung is dependent on the 3D microstructure; dysmorphia due to disease leads to tissue dysfunction. 3D XRH has already provided new insights into respiratory diseases

such as COPD [10] and idiopathic pulmonary fibrosis [11]. These studies also took advantage of the non-destructive nature of μCT to explore specific histological changes in greater detail by correlative imaging with traditional 2D histology using bright-field microscopy.

In this work we have demonstrated the enhanced utility of μCT imaging of paraffin-embedded lung samples by facilitating more automated co-registration of imaging modalities, thus leading to faster segmentation methods. This included the novel use of tissue autofluorescence as a base for automated co-registration with μCT datasets. A summary of the complete correlative workflow highlighting the steps developed in this work can be seen in figure 8, which differs from figure 1 by presenting solutions to the various issues presented by correlating 2D IF images with 3D μCT data. Applying μCT imaging settings from previous studies [2,5], we have confirmed that μCT can provide high-resolution 3D imaging of soft tissues (step 2 in figure 8). The key structural features of human lung tissue samples were revealed with isotropic voxel spacing less than 10 μm, including blood vessels, airways and alveolar structures. Since μCT is non-destructive and isotropic, datasets are free of the sectioning artefacts and distortions often seen with traditional 2D histology.

Previous correlative μCT/wax histology studies have predominantly used bright-field imaging and tinctorial/IHC staining of sections, with exceedingly time-consuming manual methods of registration and segmentation, which take several weeks or months (depending on the number of segmented images) [5,11]. Here, we added fluorescence imaging using immunostaining (step 3 in figure 8), which was very effective as a correlative imaging modality for tissue structure to complement the μCT images. This is because of the excellent contrast in the IF images between the stained tissue and the background compared to bright-field techniques. Using a far-red fluorochrome label for the secondary antibody achieved high IF image contrast and signal to noise ratio. Tissue autofluorescence from endogenous fluorophores can also be used to identify structural features in lung tissue sections [27]. However, autofluorescence is often reduced or removed as an unwanted image background [28] but, here, it was used as a crucial part of the co-registration step (step 4 in figure 8) in this correlative imaging workflow.

Sectioning artefacts present in the tissue sections prevented direct co-registration of IF and μCT images (step 4 in figure 8) and so elastic warping techniques were applied to overcome this. Manual methods of landmark-based registration and warping exist already [7,22]. However, they are all very time-consuming (several hours for accurate registration) often requiring hundreds of manually placed points per pair of images (2D section with 3D slice) to perform sufficient image warping. Overall, the manual registrations of a single pair of images can take anywhere between 20 and 60 min for an expert to complete, depending on tissue section size and quality. Other 2D–3D registration techniques between μCT, histology or electron microscopy also involve time-consuming (hours–days) manual methods [7,29,30], and in many cases did not result in the same accuracy of registration required for this study.

To overcome the issue of sectioning artefacts more rapidly, the method reported here used the SIFT feature correspondence to significantly speed up image processing time by a factor of more than 10 compared to manually identified landmarks for BigWarp. Identifying tissue features of different sizes created hierarchical SIFT correspondences. This enabled a more accurate warping as it accounted for the heterogeneity of lung tissue's microstructure, which could potentially be applied to diseased tissue with differing complexities in tissue structures in the future. Each pair of images was automatically registered and warped on average in less than 2 min, compared to the more than 20 min it would take an expert to complete manually and significantly quicker than other existing techniques with other software, while the accuracy of the automated registration matched or exceeded the existing manual techniques. Crucially, the registration process worked well despite some substantial mismatches due to tissue distortion in the paraffin wax sections which could potentially be more common in diseased tissue, like COPD which has been investigated using μCT previously [7]. Visual inspection of the images showed that the majority of the residual inaccurate registration was seen at the tissue periphery, especially along the cut edge of the tissue, where many complex distortions can occur. In this study, we focused on Ck18 immunolocalization in the airway epithelium, which was typically located away from the tissue edges, where the majority of the registration was accurate. Using a batch script also allowed whole stacks of fluorescence sections to be warped automatically without user intervention, further reducing the user effort required in co-registration and removing user bias (step 4 of figure 8). However, it is worth noting that this warping transformation is a 2D transformation applied to the IF images to match the μCT, following the previous 3D transformation of the dataset to match the cutting plane. Therefore, this technique may not account for large out-of-plane deformations caused by tissue sectioning.

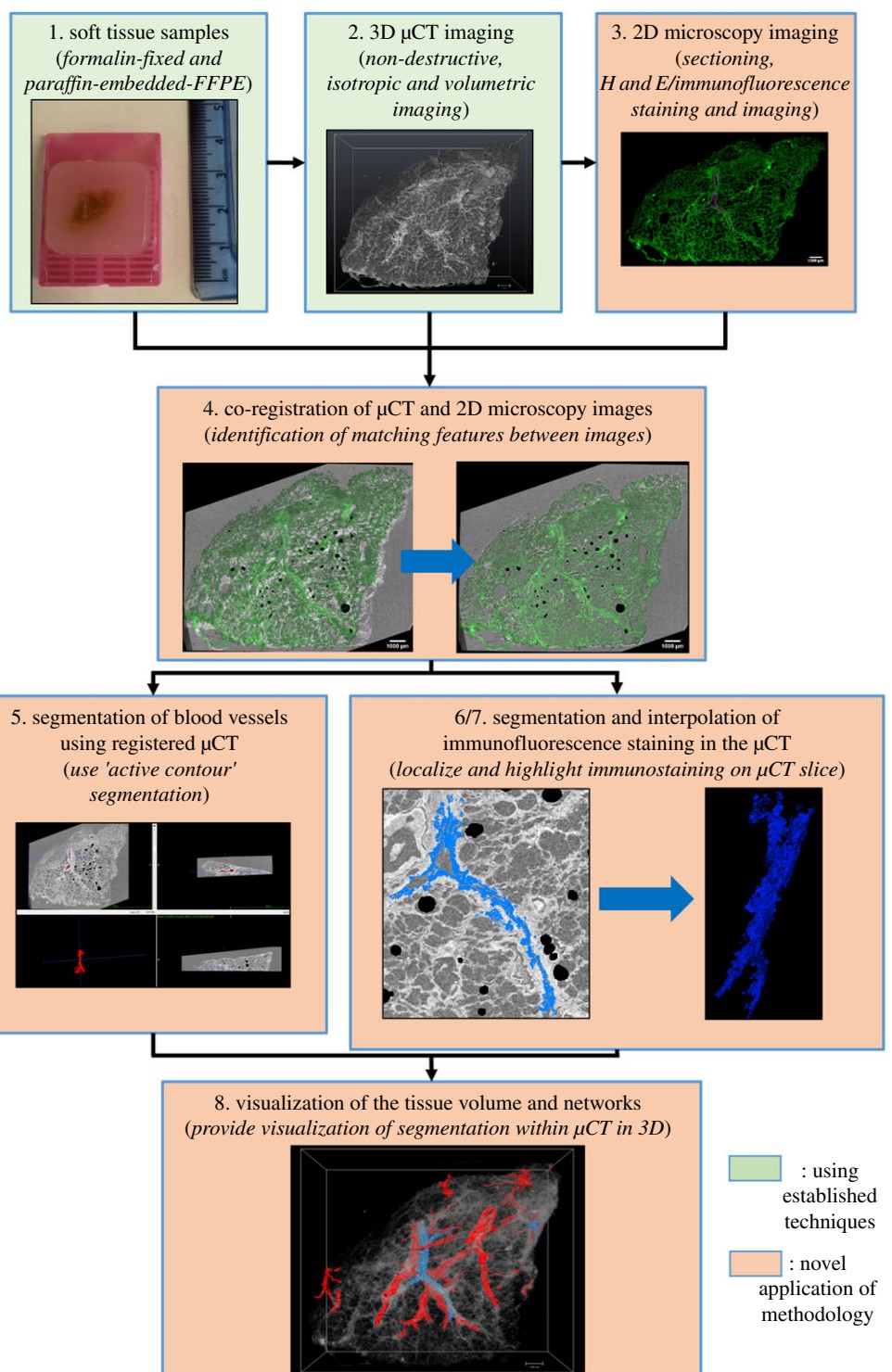

**Figure 8.** The main steps of the workflow presented in this report for automating registration and segmentation of biologically relevant features and cell types in 3D data provided by μCT. Steps highlighted in green used previously published techniques. Steps highlighted in orange were developed in this report for greater automation; these involved novel techniques or using existing techniques on new types of data. 1: Soft tissue sample preparation via FFPE. 2: Imaging of lung tissue at a high resolution (approx. 10 μm) with sufficient contrast to visualize the tissue in 3D. 3: Section tissue and perform IF staining to identify specific features not visible in μCT. 4: Automated co-registration of the specified 2D data from IF with the μCT. 5: Identify and extract the blood vessel networks from the co-registered μCT using 'active contour segmentation'. 6: Localize specific immunoreactivity, provided by IF, within the μCT and use these data for automated segmentation. 7: Digital interpolation used to 'fill the gaps' to produce 3D segmentation. 8: Bring the μCT, blood vessels and registered IF segmentation together in order to localize specific networks and features within the 3D tissue volume.

Following registration in step 4, it was possible to confirm the identity of blood vessels from H&E and autofluorescence images, which were co-registered with corresponding µCT planes, and use 'active contour segmentation' in ITK-SNAP (step 5 of figure 8). Broad structural correspondence of features in the H&E images and µCT volumes [2,7,31] helped in placing initial seed points used to expand segmentation of blood vessels (greater than or equal to 25 µm diameter) in 3D, outside the sectioned volume. Traditionally, the identification of 3D blood vessel networks would require time-consuming serial sectioning with the individual alignment of images from each tissue section with potential variations in thickness between the sections, leading to inaccurate measurement [32]. These issues are overcome by using non-destructive isotropic µCT correlated with wide-field microscopy. Previous work which has used automatic network segmentation has used samples with very high image contrast such as bone [33] or implemented complicated algorithms which are not widely applicable [34]. Using seed-point derived active contour segmentation allowed the full blood vessel network, down to approximately 25 µm diameter, to be segmented in approximately 4 h compared to several days if the complete network was segmented manually. While manual identification of the blood vessels is still necessary to provide seed points and monitor the progress of segmentation, it is still far quicker to complete than fully manual techniques. However, the active contour segmentation approach used for blood vessels could not be used to segment the airway lumen. The reasons for this are twofold. First, the airway epithelium did not have the same image contrast (grey values often less than 15 000 which is close to paraffin wax value) as the blood vessels (grey values less than 24 000) and so could not be separated by thresholding as effectively. Attempts to segment the incomplete thresholding of boundaries caused 'leaking' of segmentation out of the airway lumen into the surrounding airspaces. Second, and related to the first point, the airway walls were much thinner than the blood vessels making them harder to identify by eye and more prone to 'leaking'.

Following co-registration, the segmentation of immuno-based localization was possible (step 6 in figure 8). Since the autofluorescence and IF channels were captured on the same section, they were consequently subject to the same deformations from sectioning. Therefore, registration and the warping transformation of the autofluorescence was also applied to the IF image. Ck18 positive staining could be easily thresholded to generate a segmentation mask, which could be overlaid onto the µCT plane to highlight the areas of Ck18 localization without the need for manual segmentation. This provided specificity to µCT and 3D context to IF imaging, offering a unique perspective into tissue structure and function.

Multiple stained sections at different levels in the sectioned volume were registered to the µCT data to provide the foundations of a 3D dataset of immunolocalized features. Step 7 of figure 8 used the digital interpolation of the Ck18 localization to build 3D segmentation with minimal user inputs. Interpolation of the Ck18 staining was constrained by the tissue information in the µCT to provide an accurate segmentation mask of the slices which did not have a matching IF section registered to them. Although not as accurate as registering every serial section, interpolation meant that fewer tissue sections were needed to be stained with the Ck18 primary antibody. This would enable future work with the same tissue using these unstained sections with different primary antibodies to investigate different cell types. The ability to place specific immunoreactivity accurately within the µCT volume applies for any cellular marker as the registration was based on structures in the autofluorescence and not the IF channel. This could be used to add to the understanding of the distributions of specific cell types or any immunoreactive molecule in 3D tissues.

## 4.1. Future developments

The workflow reported in figure 8 facilitates µCT 3D segmentation while demonstrating the value of correlating 2D IF images with the µCT 3D data. Currently, this workflow has dramatically decreased the time taken to image, process and segment the µCT data from more than four to eight weeks to less than two weeks per paraffin wax block. Sample preparation and data acquisition times (µCT imaging, tissue sectioning, tissue staining and IF imaging) are now becoming comparable to the processing time. Thus, time-optimization of any of these processes could now have a significant effect on the overall time investment required for the proposed workflow. For example, further development and optimization of the µCT hardware and acquisition protocols could reduce scan time from 8 to 10 h down to less than 2 h allowing more than one process to be completed in a single work-day. One way this can be achieved using the current hardware is, for example, by reducing the number of acquired projections (under-sampled acquisition). A parametric study would be needed to quantify losses and gains, but depending on tissue size, texture and resolution, an optimized setting that maintains the contrast levels

required for segmentation, while significantly reducing acquisition time can be found. Two examples of such reduced-duration μCT acquisitions can be seen in electronic supplementary material, figure S7. Both sectioning and immunostaining can be automated and has been done in a diagnostic setting for many years [35], providing scope to further automate the workflow. The initial resampling of the μCT to the plane of histology sectioning is a manual and time-consuming process which could be automated by implementing new methods in the future [34]. The work in this study showed that the IF-based localization of Ck18 staining could be accurately registered as confirmed by tissue morphology overlaps. This confirmed its potential for further use to investigate the immunolocalization of different antibody binding sites within a 3D tissue volume, even if they are not necessarily associated with specific structural features, such as individual immune cells. Future work using this correlative imaging workflow could be used to investigate changes occurring to the 3D microstructures of lung tissue in COPD with more tissue samples processed in less time than previous studies [7]. In addition, this workflow could be used to map the distribution of specific cell types other than Ck18 within the 3D tissue volume, such as individual immune cells like macrophages which have a prominent role in COPD [36]. This could also be combined with the recent work by Metsher, which reported an interesting new development for nuclei staining of soft tissue in the μCT scan which does not affect the tissue for traditional histology [4]. This could be used to potentially relate tissue features and cell types identified in IF to nuclei localized in the 3D μCT for increased registration at a cellular level.

In conclusion, we have demonstrated a workflow for imaging human lung tissue using μCT followed by IF imaging to semi-automatically locate and identify blood vessels and immunolabelled cell types without the need for manual segmentation. These methods can be applied to the vast archives of FFPE tissues held in biobanks and enable future studies to have a higher throughput of samples for in-depth analysis of features identified by IF within the 3D data provided by μCT.

Ethics. Ethical approval was provided by the Southampton and South West Ethics Committee (number 08/H0502/32).
Data accessibility. All data connected to this paper and supplementary material are available at: doi:10.5258/SOTON/ D1421. The larger raw data are held within the research file store of the Biomedical Imaging Unit, Faculty of Medicine, University of Southampton, UK and can be provided upon reasonable request to the corresponding author.
The data are provided in the electronic supplementary material [37].
Authors' contributions. M.J.L. ran the experiments, analysed the data and wrote the first draft of the article contributing to conceptualization, data curation, methodology, software and visualization. O.L.K. developed the μCT imaging protocols and provided support in μCT image post-processing contributing to methodology, software and visualization. D.S. provided support for software in writing the Fiji macros, assisted in image analysis and methodology. A.A. collected and provided the resources of human lung tissue used in this article. O.L. provided resources for assisting with the μCT scanning and supervision on the project. I.H. provided funding acquisition as well as μCT-based resources and supervision on the project. P.S., P.L. and J.W. provided supervision, technical support and contributed to the writing of the article, as well as project conceptualization, funding acquisition, project administration and reviewing/editing the final article. All authors reviewed and approved the final manuscript.
Competing interests. The micro-computed tomography scanner (Med-X) technology development, optimized for soft tissue image contrast, was a collaborative effort between Nikon X-Tek Systems Ltd (Tring, UK) and a partnership between the μ-VIS X-ray Imaging Centre at the University of Southampton and the Biomedical Imaging Unit at Southampton General Hospital. M.J.L was partially funded by Nikon XTek Systems Ltd. O.J.L. and I.H. are Nikon X-Tek Systems Ltd employees.
Funding. This work was funded by a PhD studentship to M.J.L. provided by the Medical Research Council (grant no. 1823943) and Nikon X-Tek Systems Ltd, Tring, UK.
Acknowledgements. We would like to acknowledge the Biomedical Imaging Unit, the Histochemistry Research Unit and the μ-VIS X-ray Imaging Centre at the University of Southampton for the use of their facilities and equipment. We thank all the staff and students who have provided help and support. We acknowledge Oliver Hope for his work in training the Weka classifier for identifying air bubbles in the scans. The authors acknowledge the use of the IRIDIS High-Performance Computing Facility, and associated support services at the University of Southampton, in the completion of this work.

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
