## [Peer Review File · Royal Society Open Science]

Review History

RSOS-211067.R0 (Original submission)

Review form: Reviewer 1

Is the manuscript scientifically sound in its present form?

Yes

Are the interpretations and conclusions justified by the results?

Yes

Is the language acceptable?

Yes

Do you have any ethical concerns with this paper?

No

Have you any concerns about statistical analyses in this paper?

No

Recommendation?

Accept with minor revision (please list in comments)

Comments to the Author(s)

The authors present a workflow for registering 3d μ CT to 2d histological slices using autofluorescence, SIFT, and Bigwarp (thin-plate-spline). The resulting transformation enabled segmentation with of the μ CT with ITK-SNAPk. This work and the analyses appears sound. The data and code are made available and appear sufficient for replication.

Figure 5F and some of the introductory context need improving in my view (see below). I expect the authors will be able address my concerns by more clearly explaining the context of their work and the details of the analysis performed. Other than that, I find this to be a solid contribution and well written overall and appropriate for publication.

Figure 5F:

Are the quantitative results shown for the single section shown on other Fig 5 panels or for many sections? Please make clear over what subset of the data this analysis was done. If there is variation from section to section or between samples, consider showing the distribution of the overlap measurement across slices/samples.

I think the introduction is wanting for some more context to give the reader some points of comparison. What are the current approaches for co-registration and segmentation, and how, precisely, does the current method improve the current state of affairs? The paper currently includes broad statements that the proposed method saves time (which I believe). For segmentation, no explicit comparison is shown. For registration, this work compares to manual registration with bigwarp in section 3.3. I feel this work would be stronger if these statements could be made more concrete and precise, i.e., faster than which methods and by how much. (See the two points below)

Line 45

"However, many of the image registration methods require high amounts of manual user input to identify matching features in the images; this remains a major bottleneck in the processing of these types of images" Which image registration methods are the authors referring to (is it manual registration with bigwarp?) Are there any other automated or semi-automated approaches the authors are aware of that are reasonable choices?

Line 51:

"Segmentation of μ CT images using immunostaining for reference is currently possible but requires large amounts of time and effort to do." What approach makes it currently possible? Is it fully-manual? semi-manual?

Other minor comments / questions below:

It seems that the manual rotation step (line 134) is the only (?) step that applies a 3D transformation to the 2D IF slice. Is that correct? That is, are the SIFT+bigwarp + manual bigwarp purely 2D transformations. This should be made clear, since if so, then the approach as described would not be able to compensate for non-linear out-of-plane warping.

on line 340

"which would also work on diseased tissue with differing complexities in tissue structures." This statement feels like a conclusion, but isn't since the work presented did not explore diseased tissue. The phrasing should be softened to make it clear that it is a hypothesis/speculation, in particular the phrase "would also work" is too strong.

How many landmark pairs did SIFT produce and were given to Bigwarp? Did this vary much from section to section?

Review form: Reviewer 2

Is the manuscript scientifically sound in its present form?

Yes

Are the interpretations and conclusions justified by the results?

Yes

Is the language acceptable?

Yes

Do you have any ethical concerns with this paper?

No

Have you any concerns about statistical analyses in this paper?

No

Recommendation?

Accept as is

Comments to the Author(s)

The revised manuscript has adequately addressed reviewers comments.

Decision letter (RSOS-211067.R0)

Dear Dr Lawson

On behalf of the Editors, we are pleased to inform you that your Manuscript RSOS-211067 "Immunofluorescence-guided segmentation of three-dimensional features in micro-computed tomography datasets of human lung tissue" has been accepted for publication in Royal Society Open Science subject to minor revision in accordance with the referees' reports. Please find the referees' comments along with any feedback from the Editors below my signature.

We invite you to respond to the comments and revise your manuscript. Below the referees' and Editors' comments (where applicable) we provide additional requirements. Final acceptance of

your manuscript is dependent on these requirements being met. We provide guidance below to help you prepare your revision.

Please submit your revised manuscript and required files (see below) no later than 7 days from today's (ie 24-Sep-2021) date. Note: the ScholarOne system will 'lock' if submission of the revision is attempted 7 or more days after the deadline. If you do not think you will be able to meet this deadline please contact the editorial office immediately.

on behalf of Dr Michael Doube (Associate Editor) and Malcolm White (Subject Editor)
openscience@royalsociety.org

Associate Editor Comments to Author (Dr Michael Doube):

Comments to the Author:

Dear Dr Lawson,

Two reviewers have seen your transferred-in manuscript and they and I have a few minor suggestions that will aid in the clarity and precision of your work. Please attend to them and make a final submission.

I have a few minor comments in addition to those of the reviewer to consider while preparing your final submission:

- refer to and use consistently X-ray microtomography (XMT) rather than micro-CT, because this is a more precise name given by the technique's inventor (see Elliott & Dover 1982 "X-ray microtomography" *J Microsc* 10.1111/j.1365-2818.1982.tb00376.x)

p4 L 92 more detail on the objective lens is needed. NA, correction, etc. Also the excitation and emission details are lacking: what is the light source and what is the filter set / spectral detection, etc.? What camera is installed?

- consider 'pixel spacing' instead of 'voxel size'. Pixels are now considered to be multidimensional, and we do not have names for pixels with other combinations of spatial, time, channel, etc. dimensions. Pixels are also size-less (not little boxes / squares), but do exist as discrete samples that represent a finite piece of the object.

Reviewer comments to Author:

Reviewer: 1

Comments to the Author(s)

The authors present a workflow for registering 3d μ CT to 2d histological slices using autofluorescence, SIFT, and Bigwarp (thin-plate-spline). The resulting transformation enabled segmentation with of the μ CT with ITK-SNAPk. This work and the analyses appears sound. The data and code are made available and appear sufficient for replication.

Figure 5F and some of the introductory context need improving in my view (see below). I expect the authors will be able address my concerns by more clearly explaining the context of their work and the details of the analysis performed. Other than that, I find this to be a solid contribution and well written overall and appropriate for publication.

Figure 5F:

Are the quantitative results shown for the single section shown on other Fig 5 panels or for many sections? Please make clear over what subset of the data this analysis was done. If there is variation from section to section or between samples, consider showing the distribution of the overlap measurement across slices/samples.

I think the introduction is wanting for some more context to give the reader some points of comparison. What are the current approaches for co-registration and segmentation, and how, precisely, does the current method improve the current state of affairs? The paper currently includes broad statements that the proposed method saves time (which I believe). For segmentation, no explicit comparison is shown. For registration, this work compares to manual registration with bigwarp in section 3.3. I feel this work would be stronger if these statements could be made more concrete and precise, i.e., faster than which methods and by how much. (See the two points below)

Line 45

"However, many of the image registration methods require high amounts of manual user input to identify matching features in the images; this remains a major bottleneck in the processing of these types of images" Which image registration methods are the authors referring to (is it manual registration with bigwarp?) Are there any other automated or semi-automated approaches the authors are aware of that are reasonable choices?

Line 51:

"Segmentation of μ CT images using immunostaining for reference is currently possible but requires large amounts of time and effort to do." What approach makes it currently possible? Is it fully-manual? semi-manual?

Other minor comments / questions below:

It seems that the manual rotation step (line 134) is the only (?) step that applies a 3D transformation to the 2D IF slice. Is that correct? That is, are the SIFT+bigwarp + manual bigwarp purely 2D transformations. This should be made clear, since if so, then the approach as described would not be able to compensate for non-linear out-of-plane warping.

on line 340

"which would also work on diseased tissue with differing complexities in tissue structures." This statement feels like a conclusion, but isn't since the work presented did not explore diseased tissue. The phrasing should be softened to make it clear that it is a hypothesis/speculation, in particular the phrase "would also work" is too strong.

How many landmark pairs did SIFT produce and were given to Bigwarp? Did this vary much from section to section?

Reviewer: 2

Comments to the Author(s)

The revised manuscript has adequately addressed reviewers comments.

===PREPARING YOUR MANUSCRIPT===

===PREPARING YOUR REVISION IN SCHOLARONE===

Author's Response to Decision Letter for (RSOS-211067.R0)

See Appendix A.

Decision letter (RSOS-211067.R1)

Dear Dr Lawson,

I am pleased to inform you that your manuscript entitled "Immunofluorescence-guided segmentation of three-dimensional features in micro-computed tomography datasets of human lung tissue" is now accepted for publication in Royal Society Open Science.

on behalf of Dr Michael Doube (Associate Editor) and Malcolm White (Subject Editor)
openscience@royalsociety.org

Appendix A

Subject: Revision and resubmission of manuscript rsif-2020-0868

Dear Editor,

Thank you for accepting the manuscript for publication with minor changes. The suggestions offered by yourself and the reviewers have been immensely helpful, and we also appreciate your insightful comments on revising some parts of the paper.

I have included the reviewers' comments immediately after this letter and responded to them individually, indicating exactly how we addressed each concern or shortcoming and describing the changes we have made. The revisions have been approved by all co-authors. The changes are visible via track changes in the manuscript with all references to line numbers in the track changes version. Also the revised (clean) final manuscript with changes applied and all additional sections included has also been submitted.

Thank you for your consideration and your continued interest in our research.

Sincerely,

Matthew Lawson

3D X-ray Histology (XRH) group
Faculty of Engineering and Physical Sciences
University of Southampton

Reviewer Comments, Author Responses and Manuscript Changes

Two reviewers have seen your transferred-in manuscript and they and I have a few minor suggestions that will aid in the clarity and precision of your work. Please attend to them and make a final submission.

I have a few minor comments in addition to those of the reviewer to consider while preparing your final submission:

- refer to and use consistently X-ray microtomography (XMT) rather than micro-CT, because this is a more precise name given by the technique's inventor (see Elliott & Dover 1982 "X-ray microtomography" J Microsc 10.1111/j.1365-2818.1982.tb00376.x)

- **Comments:** *We agree that XMT is more precise and arguably an even more accurate term would be "X-ray microfocus computed tomography", which would be our preference. However, the community recognised it and refers to it as micro-CT or μ CT with most recent publications referring to it as this. Therefore we feel that it is best if kept consistent with it and use the term that's resonates best with the wider community.*
- **Changes:** *In order to make it clear for all readers the introduction has been changed on line 9 to include a mention of all the different terminology and explains for simplicity that the remainder of the manuscript will use μ CT.*

p4 L 92 more detail on the objective lens is needed. NA, correction, etc. Also the excitation and

emission details are lacking: what is the light source and what is the filterset / spectral detection, etc.? What camera is installed?

- **Changes:** *Extra details provided on the image acquisition of the fluorescence images on line 101.*

- consider 'pixel spacing' instead of 'voxel size'. Pixels are now considered to be multidimensional, and we do not have names for pixels with other combinations of spatial, time, channel, etc. dimensions. Pixels are also size-less (not little boxes / squares), but do exist as discrete samples that represent a finite piece of the object.

- **Comments:** *We agree that Voxel size contains some ambiguity as by that we actually refer to the “voxel edge” and also assume isotropicity. Your comment regarding the 3D nature of a “pixel” is interesting and might well be accepted in some communities. From our point of view we argue that by definition pixel stands for “picture element”, which makes it a 2D element. Voxel on the other hand, which stands for “volume element”, feels more appropriate term for the 3D data.*
- **Changes:** *We have changed each mention of voxel size to the less ambiguous “voxel spacing”. We hope that this is adequate at covering your point.*

Reviewer comments to Author:

Reviewer: 1

Comments to the Author(s)

The authors present a workflow for registering 3d μ CT to 2d histological slices using autofluorescence, SIFT, and Bigwarp (thin-plate-spline). The resulting transformation enabled segmentation with of the μ CT with ITK-SNAP. This work and the analyses appears sound. The data and code are made available and appear sufficient for replication.

Figure 5F and some of the introductory context need improving in my view (see below). I expect the authors will be able address my concerns by more clearly explaining the context of their work and the details of the analysis performed. Other than that, I find this to be a solid contribution and well written overall and appropriate for publication.

Figure 5F:

Are the quantitative results shown for the single section shown on other Fig 5 panels or for many sections? Please make clear over what subset of the data this analysis was done. If there is variation from section to section or between samples, consider showing the distribution of the overlap measurement across slices/samples.

- **Comments:** You are correct that this was for the single slice. This has been changed to cover all pairs of images analysed.
- **Changes:** Figure and reporting of results in text changed to average of 22 pairs of images (74.1 \pm 4.2) (line263-270). Also added to methods section stating analysis was done on multiple images (line 179).

I think the introduction is wanting for some more context to give the reader some points of

comparison. What are the current approaches for co-registration and segmentation, and how, precisely, does the current method improve the current state of affairs? The paper currently includes broad statements that the proposed method saves time (which I believe). For segmentation, no explicit comparison is shown. For registration, this work compares to manual registration with bigwarp in section 3.3. I feel this work would be stronger if these statements could be made more concrete and precise, i.e., faster than which methods and by how much. (See the two points below)

- **Comments:** Worked through the manuscript to make the time comparisons more precise between previous methods and the one reported here. Also added some more context for the purpose of the study in the introduction section (line 62).

Line 45

"However, many of the image registration methods require high amounts of manual user input to identify matching features in the images; this remains a major bottleneck in the processing of these types of images" Which image registration methods are the authors referring to (is it manual registration with bigwarp?) Are there any other automated or semi-automated approaches the authors are aware of that are reasonable choices?

- **Changes:** Added to the sentence to include specific references to some of the software currently used in the community for image registration including open source (ImageJ and ec-CLEM and commercial (Avizo and Zen) (line 52).

Line 51:

"Segmentation of μ CT images using immunostaining for reference is currently possible but requires large amounts of time and effort to do." What approach makes it currently possible? Is it fully-manual? semi-manual?

- **Changes:** *Added comments that the majority of previous methods for segmenting immunostaining is manual with references to recent literature.* (line 60)

Other minor comments / questions below:

It seems that the manual rotation step (line 134) is the only (?) step that applies a 3D transformation to the 2D IF slice. Is that correct? That is, are the SIFT+bigwarp + manual bigwarp purely 2D transformations. This should be made clear, since if so, then the approach as described would not be able to compensate for non-linear out-of-plane warping.

- **Comments:** *The previous step of resampling the μ CT data to the histology is a 3D transform to match the histology but you are correct that there is no further 3D transformation to the IF images.*
- **Changes:** Added a point in the discussion to highlight this and making sure the reader understands that the warping is a 2D transformation of the IF images to match the μ CT plane. (Line 380)

on line 340

"which would also work on diseased tissue with differing complexities in tissue structures." This statement feels like a conclusion, but isn't since the work presented did not explore diseased tissue. The phrasing should be softened to make it clear that it is a hypothesis/speculation, in particular the phrase "would also work" is too strong.

- **Changes:** *Changed phrasing to make the statement softer as a potential future application of the technique to be explored (line 365).*

How many landmark pairs did SIFT produce and were given to Bigwarp? Did this vary much from section to section?

- **Comments:** *Manually it was approximately 50 landmarks whereas SIFT (after removing duplicates produced between 50-200 depending on the size of the tissue section.*
- **Changes:** *Added these number in on line 267 and commented on the variation in line 269.*

Reviewer: 2

Comments to the Author(s)

The revised manuscript has adequately addressed reviewers comments.